# Asteroid shower on the Earth-Moon system immediately before the Cryogenian period revealed by KAGUYA

Kentaro Terada 1✉, Tomokatsu Morota 2,3 & Mami Kato3,4

Meteoroid bombardment of the Earth-Moon system must have caused catastrophic damage to the terrestrial ecosphere. However, ancient meteoroid impacts and their relations to environmental changes are not well understood because of erosion and/or resurfacing processes on Earth. Here, we investigate the formation ages of 59 lunar craters with fresh morphologies and diameters greater than approximately 20 km and first find that 8 of 59 craters were formed simultaneously. Considering the radiometric ages of ejecta from Copernicus crater and impact glass spherules from various Apollo landing sites, we conclude that sporadic meteoroid bombardment occurred across the whole Moon at approximately 800 Ma. Based on crater scaling laws and collision probabilities with the Earth and Moon, we suggest that at least $(4-5) \times 10^{16}$ kg of meteoroids, approximately 30–60 times more than the Chicxulub impact, must have plunged into the Earth-Moon system immediately before the Cryogenian, which was an era of great environmental changes.

[1] Department of Earth and Space Science, Osaka University, Toyonaka 560-0043, Japan. [2] Department of Earth and Planetary Science, The University of Tokyo, Bunkyoku 113-0033, Japan. [3] Department of Earth and Planetary Sciences, Nagoya University, Nagoya 464-8601, Japan. [4] Present address: MEISEI ELECTRIC CO., LTD, Isesaki 372-8585, Japan. ✉email: terada@ess.sci.osaka-u.ac.jp

Understanding meteoroid bombardment of the Earth system is an issue of both great scientific interest and practical importance because impacts are potentially hazardous to the Earth. Since the 541 Ma Cambrian biodiversity explosion, mass extinction events have occurred at least five times (the so-called Big five events)[1], and extra-terrestrial impacts are considered a potential cause of some of them (e.g., Late Triassic[2] and Cretaceous-Palaeogene extinctions[3]), competing with the flood basalt eruption-related hypotheses[4].

After the first discovery of fossil L-chondrites in Ordovician limestones in Sweden[5], abundant L-chondrites, meteorite-tracing chromite grains and iridium enrichment have been found in Sweden, England, Scotland, China, and Russia in rocks whose stratigraphic ages are 470 ~ 480 million years (Ma)[6,7]. Moreover, several large terrestrial craters in the Northern Hemisphere have been found to have radiometric ages of approximately 430 ~ 470 Ma[8]. Further, approximately two-thirds of ordinary L-chondrites are known to be heavily shocked and degassed, with $^{39}$Ar–$^{40}$Ar ages near 470 Ma[9]. Therefore, it is generally considered that the L-chondrite parent body suffered a major impact approximately 470 million years ago and was catastrophically disrupted, causing a very large meteoroid shower on Earth for several million years[10]. Recently, Schmitz et al.[11] suggested that the extraordinary amounts of dust during >2 Myr cooled the Earth and triggered Ordovician icehouse conditions, sea-level fall, and major faunal turnovers related to the Great Ordovician Biodiversification Event. However, to date, other ancient meteoroid impacts and their relations to environmental changes have not been well understood because of erosion and/or resurfacing processes on Earth.

Another way to reveal ancient meteoroid impacts on Earth is to investigate the lunar crater record because there is less weathering and erosion on the Moon. The lunar orbiter, Kaguya provides us a new insight that disruption of asteroid had occurred and formed the several craters larger than 20 km simultaneously on the Moon approximately 800 Ma. Based on crater scaling laws and collision probabilities with the Earth and Moon, at least $(4–5) \times 10^{16}$ kg of meteoroids, approximately 30–60 times more than the Chicxulub impact, must have struck the Earth, immediately before the Cryogenian, which was an era of great environmental and biological changes.

## Results and discussion

**Observation of crater size-frequency distribution on the Moon**. Crater size-frequency distribution measurement is a well-established technique to derive relative and absolute ages of planetary surfaces[12]; thus, the density of 0.1–1 km-diameter craters in the ejecta of a large crater (>20 km) potentially gives the formation age of the large crater itself. In this study, we investigate the formation age distribution of 59 lunar craters with fresh morphology and diameters larger than approximately 20 km (Fig. 1) using the software tool craterstats[13]. Typical craters and small craters counted for their crater size-frequency distribution measurement are shown in Fig. 2. Here, we select and investigate the regions where there is no pond (impact melt region) to avoid the target property effects that may cause craters formed in impact melts to be smaller than those in ejecta, as discussed by[13].

First, we estimate the formation ages of individual craters using the conventional constant flux model over 3 billion years (another model is discussed later). Table 1 summarizes the estimated formation ages of 59 lunar craters with fresh morphology. The remarkable new finding is that eight of 59 craters, including Copernicus, are concentrated at approximately 660 Ma, and the weighted mean is 658 ± 16 Ma (Fig. 3b). The spatial distribution of these craters seems to be slightly concentrated in the equatorial plane, but there is no significant difference between the far and near sides (Fig. 1).

To evaluate the probability of the observed concentration of crater ages, we perform a simple test using Monte Carlo simulation. We assume that craters are created with uniform probability within an age range from 3.0 Ga to 0 Ga and compute the ages of the 59 craters using a uniformly distributed pseudo-random number. The procedure is iterated 100,000 times. The results show that the possibility that seven of the 59 craters formed at the same time (for 50 Ma from 630 Ma to 680 Ma) by chance is 0.69%, where the 54S161E crater (747 ± 92 Ma) is masked because it is an obvious outlier with large uncertainties (if the 4S161E crater is included, the possibility that eight of the 59 craters formed during a 100 Ma interval by chance is 7%). From these considerations, we conclude that sporadic meteorite bombardment occurred across the whole Moon, possibly due to the disruption of asteroids, analogous to the Ordovician meteorite shower.

According to Shibaike et al.[14], the mass of an impactor can be estimated from the density of the impactor, the density of the crust, the velocity of the impactor and the diameter of the crater (see "Methods" for details). Assuming a density of near-Earth asteroids (1.29 g cm$^{-3}$ for C-type asteroid Ryugu[15], 1.9 g cm$^{-3}$ for S-type asteroid Itokawa[16] and 2.7 g cm$^{-3}$ for S-type asteroid Eros[17]) and a relative velocity of 20 km sec$^{-1}$ of Earth-crossing asteroids to the Moon[18,19], the masses and sizes of the impactors for eight lunar craters formed at 660 Ma are calibrated (see Table 2). As a result, the total mass of the asteroid shower on the Moon is estimated to be $(1.3–1.6) \times 10^{15}$ kg, corresponding to an impactor 10–13 km in diameter.

**Consideration of absolute age of clustered craters**. To date, the lunar impact history has been well investigated based on lunar impact glasses collected by the Apollo/Luna missions and/or lunar meteorites. The age of Copernicus crater is generally taken as ~800 Ma based on both crater chronology[20] and the radiometric dating of 12033 brecciated soil, which is considered to consist of ejecta from Copernicus crater[21,22]. The discrepancy in crater age between 800 Ma[20] and 660 Ma in this study is due to the difference in the selected area to be counted as follows. In fact, we investigated the same area as Hiesinger et al.[20], giving an age of 797 ± 52 Ma, whereas Hiesinger et al.[20] also reported ages of 678 ± 81 Ma for the area observed by KAGUYA CE1 and 678 ± 270 Ma for the Copernicus ray. In addition, we investigate other areas around Copernicus crater (floor, ejecta area and melt region near central peak), giving ca. 660 Ma (Fig. 4). All of these results mean that there is no discrepancy in counting and calibration between previous work[20] and this study. We also realize that the selected area with an age of 800 Ma is close to the centre of Copernicus crater and tends to be affected by secondary craters, so a counting method yielding a younger age of 660 Ma is correct for Copernicus crater. Note that the most important of the new findings is that eight craters, including Copernicus, show identical relative ages based on a constant flux model.

On the other hand, the absolute age of Copernicus crater is considered to be 800 Ma based on the radiometric ages of 12033 brecciated soil collected from the ejecta of Copernicus crater[21,22]. In addition, Zellner et al. (2015)[23] recently reported that $^{40}$Ar/$^{39}$Ar data for impact spherules from Apollo 12, 14, 16, and 17 samples show an 800 Ma spike, similar to that of the 12033 breccia, and concluded that there must have been a transient increase in the global lunar impact flux at 800 Ma other than Copernicus crater, in the context of diverse compositional ranges and sample locations of impact glass spherules. Such geochemical observations of simultaneous global lunar impacts

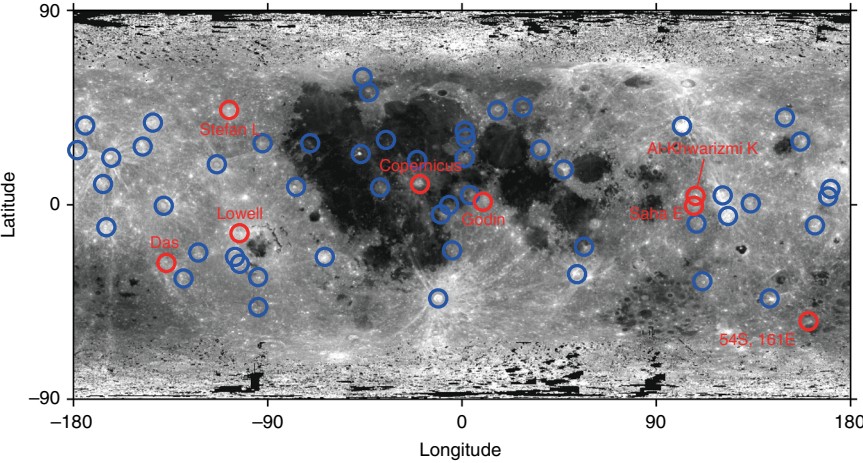

**Fig. 1 The locations of the 59 investigated craters with fresh morphologies and diameters larger than ~20 km.** The locations of the 59 investigated lunar craters with fresh morphologies and diameters larger than approximately 20 km are shown. The craters with ages the same as that of Copernicus crater are indicated by red circles.

recorded in Apollo samples well match the coincidence of (at least) eight crater formations derived from our observations, of which the probability is 0.69%. From these considerations, we infer that these two observations must be related to each other and newly propose a constant with a spike model of ~800 Ma instead of a conventional constant model (details of a new model are discussed in the next section).

This scenario in which sporadic asteroid showers did not occur at 660 Ma but at 800 Ma is also supported by recent LRO observations and/or numerical simulations of the asteroid families as follows[24–27]. Based on the temperature of large impact ejecta with crater sizes larger than 10 km in diameter, Mazrouei et al.[24] reported that there is no evidence of a sporadic peak at approximately 660 Ma, although they found that the production rate of D ≥ 10 km lunar craters was 2–3 times higher over the last ~290 Myr. However, the age of 800 Ma is very close to the limit of resolution shown in Fig. 3 in Mazrouei et al.[24], so there is no contradiction with the 800 Ma spike model. Furthermore, numerical simulation of the orbits of asteroid families also provides crucial chronological information about the impact flux to the inner solar system. A recent investigation of the dynamics of the asteroid family and the best available Yarkovsky measurement suggest[25] that the breakup of 847 Agnia (669–1003 Ma) and/or 480 Hansa (763–950 Ma) was related to the sporadic asteroid shower at 660 Ma. However, it is known that these families are not sizable enough and/or not well enough positioned to produce the sporadic asteroid shower, including Copernicus crater with a diameter of 93 km, for which impactor is expected to be 10 km in diameter (Table 2). Moreover, the Agnia family is located near the 5:2 resonance, where the probability of a projectile hitting the Moon is very low (~10⁻⁵)[26]. The Hansa family also has difficulty producing an impactor of 10 km for Copernicus because it is located at high inclinations near the 3:1 and 8:3 resonances. However, the Eulalia family, whose age is 830 [+370, −100] Ma, could potentially have produced an impact spike at ~800 Ma. According to Bottke et al.[27], when the parent body of Eulalia was disrupted, a large share of the sizable family was directly injected into the 3:1 resonance at low inclinations. This disruption certainly could have produced an impact spike on terrestrial planets and/or their satellites inside the asteroid belt. Interestingly, the Eulalia family is a carbonaceous chondrite family and is considered to be the parent body of near-Earth C-type asteroids, such as Bennu and Ryugu[27]. Such an asteroid shower must have contaminated the lunar surface with volatile elements. This scenario is harmonized with (i) the observation of

H₂O in Copernicus crater that may reflect retention of volatiles from hydrous impactors[28]; (ii) the scenario that may have been formed by a cometary nucleus, 4 km in diameter based on geochemistry of the 12033 breccia[29]; and (iii) recent KAGUYA remote-sensing observation of persistent C⁺ emitted from the whole Moon, which is significantly larger than influxes due to solar wind and/or current micrometeoroid accretion[30] and suggests that the lunar surface might have been contaminated by volatile-rich impactors in the past.

**Crater age distribution by the 800 Ma spike model.** It is obvious that the break-up of large asteroids increases not only the large (>20 km) crater production rate but also the small (0.1–1 km) crater production rate. From these considerations, we propose the new simplest model: a constant flux with a spike between 830 and 800 Ma for small craters (0.1–1 km), as shown in Fig. 4 (hereafter, we call this concept the 800 Ma spike model). The basic idea is that the crater counting age of Copernicus crater must be identical to the radiometric age of 800 Ma and that the fluxes before and after the sporadic spike at ~800 Ma were constant. Although the duration time of the spike is slightly uncertain, we assume that this duration was 30 Ma (from 830 Ma to 800 Ma), based on the break-up age of Eulalia[26] and the radiometric age of Copernicus crater[21,22] and/or the deviation of eight clustered ages (658 ± 16 Ma) for a constant model. Figure 5 illustrates the difference between the conventional constant flux model and the 800 Ma spike model. The constant flux of the new spike model is 75% (=663 Ma/800 Ma) of the conventional constant flux model, and the flux between 800 Ma and 830 Ma is 23 times higher than that in other eras (Fig. 5a) to ensure that the total crater production over 3 billion years is identical for both models, as shown in Fig. 5b).

As a result, the modified age distribution shows that 16 of the 59 craters coincide with that of Copernicus crater within the analytical certainties (Fig. 3b, Table 2), although the large (>20 km) crater production rate and the small (0.1–1 km) crater production rate might be coupled in this model. However, the estimated total masses are not significantly changed ((1.3–1.6) × 10¹⁵ kg for the constant flux model and (1.8–2.3) × 10¹⁵ kg for the 800 Ma spike model) because Copernicus crater is dominant among the eight coincident craters (by the constant model) and the 17 coincident craters (by the spike model). Therefore, the latter discussion on the total mass estimation of the impactor is not affected by the choice of a flux model with/without the spike.

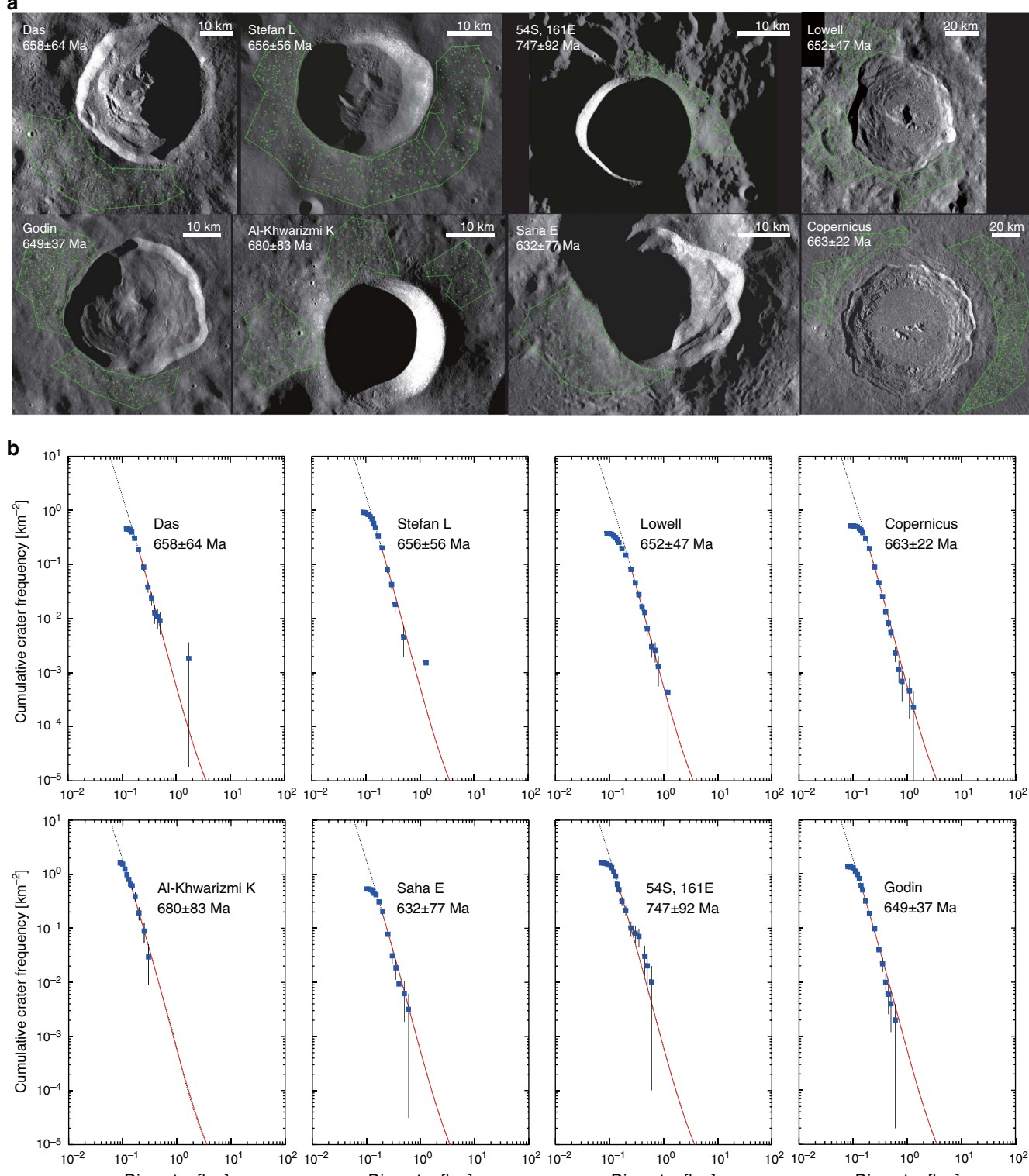

**Fig. 2 The Terrain Camera images of individual craters and their cumulative size-frequency distributions, for which ages are identical to that of Copernicus crater. a** Mosaics of the Terrain Camera images of individual craters shown in simple cylindrical map projection and **b** the cumulative size-frequency distributions, from which ages are identical to that of Copernicus crater. The small craters around the central main crater are counted for the determination of the central crater itself. **b** Shows their cumulative size-frequency distributions. Error bars are estimated by the formula $(n \pm n^{1/2})/A$, where $n$ is the cumulative number of craters and $A$ is the counted area.

| Table 1 Size. Location and model age of lunar craters (>~15 km) with fresh morphology. | | | | | | | | | |
|---|---|---|---|---|---|---|---|---|---|
| | | | Diameter | | | Constant Model | | 800 Ma spike model | |
| Name | Location | ID | {km} | Latitude | Longitude | Age [Myr] | Error | Age [Myr] | Error |
| Giordano Bruno | Far | GoB | 21.3 | 35.95 | 102.88 | 6 | 1 | 7 | 1 |
| Moore F | Far | MoF | 23.7 | 37.28 | 185.04 | 42 | 10 | 51 | 12 |
| Byrgius A | Near | ByA | 18.7 | −24.55 | 296.19 | 46 | 2 | 56 | 2 |
| Tycho | Near | Tyc | 83.2 | −43.32 | 348.80 | 58 | 3 | 70 | 4 |
| 43S143E | Far | 43S | 14.7 | −43.88 | 143.30 | 68 | 10 | 82 | 12 |
| Larmor Q | Far | LrQ | 25.5 | 24.79 | 181.40 | 82 | 11 | 99 | 13 |
| Necho | Far | Nec | 31.2 | −5.23 | 123.27 | 82 | 4 | 99 | 5 |
| Proclus | Near | Pro | 27.2 | 16.08 | 46.93 | 104 | 10 | 125 | 12 |
| Aristarchus | Near | Ara | 40.0 | 23.73 | 312.51 | 132 | 11 | 159 | 13 |
| Jacson | Far | Jac | 70.9 | 22.02 | 196.62 | 144 | 19 | 174 | 23 |
| OlberA | Near | OlA | 41.8 | 8.09 | 282.32 | 185 | 15 | 223 | 18 |
| PetaviusB | Near | PtB | 33.6 | −19.96 | 57.04 | 227 | 34 | 274 | 41 |
| Ohm | Far | Ohm | 63.9 | 18.25 | 246.25 | 245 | 17 | 296 | 21 |
| p.o. Van Newman F | Far | VNF | 32.4 | 40.86 | 149.95 | 279 | 40 | 337 | 48 |
| Lalande | Near | Lal | 23.3 | −4.47 | 351.39 | 300 | 24 | 362 | 29 |
| Kepler | Near | Kep | 29.9 | 8.11 | 322.01 | 453 | 26 | 547 | 31 |
| Crookes | Far | Cro | 48.8 | −10.35 | 194.92 | 479 | 32 | 578 | 39 |
| King | Far | Kin | 76.5 | 4.93 | 120.51 | 536 | 38 | 647 | 46 |
| Saha E | Far | ShE | 29.4 | −0.32 | 108.04 | 632 | 77 | 763 | 93 |
| Godin | Near | God | 35.1 | 1.82 | 10.15 | 649 | 37 | 783 | 45 |
| Lowell | Far | Low | 65.6 | −12.90 | 256.56 | 652 | 47 | 787 | 57 |
| Stefan L | Far | StL | 26.0 | 44.36 | 251.91 | 656 | 56 | 792 | 68 |
| Das | Far | Das | 36.6 | −26.52 | 222.93 | 658 | 64 | 794 | 77 |
| Copernicus | Near | Cop | 93.1 | 9.62 | 339.93 | 663 | 22 | 800 | 27 |
| Al-Khwarizmi K | Far | AlK | 22.5 | 4.48 | 108.16 | 680 | 83 | 821 | 100 |
| 54S161E | Far | 54S | 19.7 | −53.93 | 161.18 | 747 | 92 | 804 | 99 |
| Stevinus | Near | Ste | 70.3 | −32.53 | 54.14 | 766 | 40 | 805 | 42 |
| Klute W | Far | KlW | 30.2 | 37.98 | 216.71 | 768 | 80 | 805 | 84 |
| Zhukovsky Z | Far | ZuZ | 33.9 | 9.82 | 192.79 | 889 | 130 | 812 | 119 |
| Golitsyn | Far | Gol | 37.2 | −25.14 | 254.82 | 1080 | 120 | 822 | 91 |
| Milne N | Far | MiN | 32.9 | −35.83 | 111.33 | 1080 | 150 | 822 | 114 |
| Guthnic | Far | Gut | 36.2 | −47.78 | 265.98 | 1100 | 80 | 823 | 60 |
| Steno Q | Far | StQ | 32.0 | 29.04 | 157.73 | 1140 | 140 | 825 | 101 |
| Robinson | Near | Rob | 24.0 | 59.05 | 313.99 | 1160 | 190 | 826 | 135 |
| Harpalus | Near | Har | 40.4 | 52.73 | 316.49 | 1230 | 60 | 830 | 40 |
| Vavilov | Far | Vav | 96.6 | −0.90 | 221.22 | 1260 | 70 | 859 | 48 |
| Romer | Near | Rom | 41.7 | 25.38 | 36.42 | 1290 | 110 | 895 | 76 |
| Aristillus | Near | Arl | 54.5 | 33.87 | 1.22 | 1330 | 110 | 944 | 78 |
| Plante | Far | Pla | 35.3 | −10.23 | 163.29 | 1370 | 180 | 992 | 130 |
| Laue G | Far | LuG | 29.8 | 27.91 | 266.70 | 1430 | 110 | 1064 | 82 |
| Burg | Near | Bur | 38.6 | 45.07 | 28.23 | 1460 | 120 | 1100 | 90 |
| Conon | Near | Con | 20.8 | 21.65 | 1.96 | 1610 | 200 | 1281 | 159 |
| Delisle | Near | Del | 25.5 | 29.97 | 325.32 | 1610 | 170 | 1281 | 135 |
| Autolycus | Near | Aut | 38.9 | 30.69 | 1.47 | 1660 | 260 | 1342 | 210 |
| Triesnecker | Near | Tri | 25.3 | 4.18 | 3.60 | 1670 | 170 | 1354 | 138 |
| Thebit A | Near | ThA | 20.5 | −21.60 | 355.10 | 1760 | 170 | 1462 | 141 |
| Eudoxus | Near | Eud | 67.1 | 44.27 | 16.38 | 1790 | 120 | 1499 | 100 |
| Focas | Far | Foc | 22.0 | −33.71 | 266.10 | 1800 | 200 | 1511 | 168 |
| Coriolis Y | Far | CrY | 31.2 | 3.55 | 170.98 | 1870 | 300 | 1595 | 256 |
| Phtheas | Near | Pyt | 19.7 | 20.55 | 339.41 | 1950 | 210 | 1692 | 182 |
| Green M | Far | GrM | 34.7 | 0.36 | 133.13 | 1970 | 210 | 1716 | 183 |
| Golitsyn J | Far | GlJ | 19.0 | −27.69 | 256.79 | 1970 | 320 | 1716 | 279 |
| Gerasimovich D | Far | GeD | 25.9 | −22.45 | 237.98 | 2000 | 440 | 1752 | 385 |
| Mosting | Near | Mos | 24.9 | −0.71 | 354.14 | 2200 | 170 | 1993 | 154 |
| Briggs B | Near | BrB | 24.8 | 28.15 | 289.12 | 2330 | 220 | 2150 | 203 |
| Dufay B | Far | DuB | 21.6 | 8.38 | 171.18 | 2370 | 270 | 2198 | 250 |
| Pasteur D | Far | PsD | 39.3 | −9.05 | 109.12 | 2380 | 290 | 2210 | 269 |
| Joule T | Far | Jou | 38.1 | 27.50 | 211.93 | 2390 | 350 | 2223 | 325 |
| 34S 130W | Far | 34S | 19.4 | −34.27 | 230.35 | 2470 | 890 | 2319 | 836 |

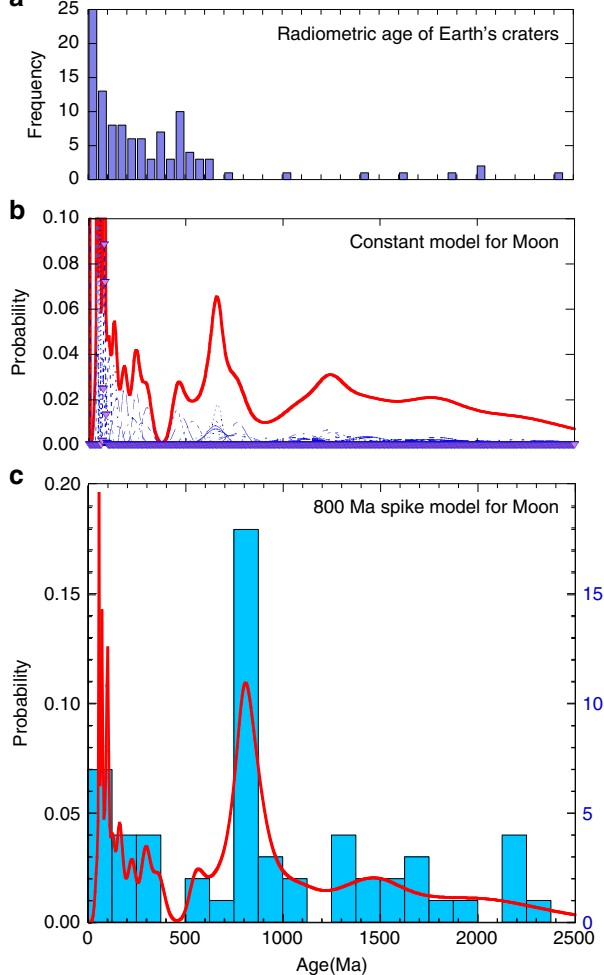

**Fig. 3 The age distributions of terrestrial craters and lunar craters. a** is a histogram of radiometric ages of terrestrial craters, **b** is the relative probability of 59 crater ages investigated based on the conventional constant flux, and **c** shows the relative probability and histogram of 59 crater ages based on the 800 Ma spike model.

Moreover, it should also be noted that the slopes of the lines <300 Ma in both models in Fig. 6 are gentler than those of other eras, which is quite consistent with the previous study that the production rate of lunar craters (>10 km) has been 2–3 times higher over the last ~290 Myr, which is derived from an independent approach based on the temperature of large impact ejecta[24].

Recent observations by Chandrayaan-1[31] and LADEE[32] suggest that an active water cycle exists on the Moon and that hydrated soil is present under the desiccated soil layer of several centimetres over the Moon surface. In particular, Li and Milliken[31] discuss that the Copernicus crater exhibits high water content that may reflect the retention of volatiles from hydrous impactors according to the Moon Mineralogy Mapper. In addition, we recently found that $C^+$ is persistently emitted from the whole of the Moon as detected by the lunar orbiter KAGUYA and that this flux is significantly larger than the influx estimated from solar wind and/or current micrometeoroid accretion[30]. These observations suggest that volatile elements are ubiquitous over the lunar surface and that the Moon is currently in the process of losing volatiles (water, carbon, etc.), although when and how the surface of the Moon attained and/or retained such volatiles has been enigmatic. Assuming the CI-like chemical

composition (a few wt.% of carbon and $H_2O$), our scenario predicts that a C-type asteroid shower at 800 Ma must have supplied ~$10^{14}$ kg of carbon and $H_2O$ to the lunar surface. This new paradigm undoubtedly should place new constraints on the history of lunar volatiles.

**Estimated flux on the Earth**. Since the Earth–Moon system has been co-evolving over 4.5 billion years, this new finding provides us with crucial insight into the Earth–Moon system because asteroid showers must have occurred not only on the Moon but also on the Earth. Based on the probability ratio of collisions with the Earth and the Moon of 23:1[33], we conclude that a mass of $(4–5) \times 10^{16}$ kg (corresponding to a diameter of ~30–40 km and ~30–60 times greater in mass than the Chicxulub asteroid impactor[34]) must have collided successively on the Earth at ~800 Ma, i.e., immediately before the Cryogenian (720–635 Ma), which was an era of great environmental and biological changes[35,36]. To date, however, no direct geological evidence of a large-scale impact in the Neoproterozoic has been found[37]. Moreover, remarkable Ir concentrations as well as other platinum-group elements (PGE) anomalies, such as those at the K–T boundary[34], have not been found, although only the Marinoan glaciation (650–635 Ma) is characterized by increased concentrations of Ir[38,39]. The straightforward interpretation is that the following large-scale Neoproterozoic glaciations, the so-called Snowball Earth (that is, the Kaigas-Sturtian glaciation from 730 to 700 Ma and the Marinoan glaciation) and/or their deglaciation processes, might have erased a significant part of the earlier geological and/or geochemical history, as discussed by[40].

To date, the impact history and subsequent effects on the environment in the Neoproterozoic and Cryogenic have not been understood because terrestrial craters are not well preserved due to erosion. Koeberl and Ivanov[41] recently discussed the mechanical effects of one impact of an asteroid 5–10 km in diameter on the Snowball Earth environment, suggesting that the products of impact (mainly water vapour) could be quickly distributed over a substantial part of the globe, influencing the global circulation (e.g., facilitating cloud formation), because one impact cratering event (shock waves and impact crater formation) might produce much dust that entered the atmosphere and might have caused albedo changes. Recently, Schmitz et al.[11] noted that the Ordovician meteorite shower should have triggered the mid-Ordovician ice age based on the sizes of the remaining terrestrial craters. Thus, large asteroid showers should influence the global ecosphere in some ways, although mechanisms are not well realized because of the unknown characteristics of the dust, e.g., size, albedo, mineralogy, and chemical composition.

Interestingly, Reinhard et al.[42] found that the average $P$ content of late Tonian samples is more than four times greater than that of pre-Cryogenian samples and noted that a fundamental shift in the phosphorus cycle may have occurred during the late Proterozoic Eon after 800 Ma (until 635 Ma). Our new finding suggests that ~$10^{14}$ kg of extra-terrestrial $P$ should have accreted across the Earth assuming CI chondrite composition ($P = 0.1$ wt. %) at 800 Ma, which is one order of magnitude higher than the total $P$ amount of the modern sea (assuming that the volume of modern seas is $13.5 \times 10^8$ km$^3$ and the concentration of $P$ is ~3 μg/litre). In general, large-scale changes in marine biogeochemical cycles are undoubtedly forced by tectonic and magmatic processes and chemical weathering of the continental crust, but our new finding suggests that the flux of extra-terrestrial bioavailable elements might also have influenced marine biogeochemical cycles[42], marine redox states[43], severe perturbations to Earth's climate system[44], and the emergence of animals[45,46]. Thus, lunar crater chronology provides new insight

**Table 2 The estimated mass and size of impactors which formed the craters with the Copernicus formation simultaneously.**

| Crater Moon | Location | Diameter of crater (km) | Constant model Age (Ma) | Spike model Age (Ma) | Mass of impactor (kg) For the case of C-type | Radius of impactor (km) (1.3 g cm$^{-3}$) | Mass of impactor (kg) For the case of S-type | Radius of impactor (km) (2.7 g cm$^{-3}$) |
|---|---|---|---|---|---|---|---|---|
| Saha E[a,b] | Far | 29.4 | 632 ± 77 | 763 ± 93 | 1.4E + 13 | 1.4 | 1.2E + 13 | 1.0 |
| Godin[a,b] | Near | 35.1 | 649 ± 37 | 783 ± 45 | 2.9E + 13 | 1.7 | 2.3E + 13 | 1.3 |
| Lowell[a,b] | Far | 65.6 | 652 ± 47 | 787 ± 57 | 3.2E + 14 | 3.9 | 2.6E + 14 | 2.8 |
| Stefan L[a,b] | Far | 26.0 | 656 ± 56 | 792 ± 68 | 9.0E + 12 | 1.2 | 7.3E + 12 | 0.9 |
| Das[a,b] | Far | 36.6 | 658 ± 64 | 794 ± 77 | 3.4E + 13 | 1.8 | 2.7E + 13 | 1.3 |
| Copernicus | Near | 93.1 | 663 ± 22 | 800 ± 27 | 1.2E + 15 | 6.1 | 1.0E + 15 | 4.4 |
| Al-Khwarizmi K[a,b] | Far | 22.5 | 680 ± 83 | 821 ± 100 | 5.2E + 12 | 1.0 | 4.2E + 12 | 0.7 |
| 54S161E[a,b] | Far | 19.7 | 747 ± 92 | 804 ± 99 | 3.1E + 12 | 0.8 | 2.5E + 12 | 0.6 |
| Stevinus[b] | Near | 70.3 | 766 ± 40 | 805 ± 42 | 4.1E + 14 | 4.2 | 3.4E + 14 | 3.1 |
| Klute W[b] | Far | 30.2 | 768 ± 80 | 805 ± 84 | 1.6E + 13 | 1.4 | 1.3E + 13 | 1.0 |
| Zhukovsky Z[b] | Far | 33.9 | 889 ± 130 | 812 ± 119 | 2.5E + 13 | 1.7 | 2.0E + 13 | 1.2 |
| Golitsyn[b] | Far | 37.2 | 1080 ± 120 | 822 ± 91 | 3.6E + 13 | 1.9 | 2.9E + 13 | 1.4 |
| Milne N[b] | Far | 32.9 | 1080 ± 150 | 822 ± 114 | 2.2E + 13 | 1.6 | 1.8E + 13 | 1.2 |
| Guthnic[b] | Far | 36.2 | 1100 ± 80 | 823 ± 60 | 3.2E + 13 | 1.8 | 2.6E + 13 | 1.3 |
| Steno Q[b] | Far | 32.0 | 1140 ± 140 | 825 ± 101 | 2.0E + 13 | 1.5 | 1.6E + 13 | 1.1 |
| Robinson[b] | Near | 24.0 | 1160 ± 190 | 826 ± 135 | 6.6E + 12 | 1.1 | 5.4E + 12 | 0.8 |
| Harpalus[b] | Near | 40.4 | 1230 ± 60 | 830 ± 40 | 4.9E + 13 | 2.1 | 4.0E + 13 | 1.5 |
| Total mass of impactors (based on the constant model) | | | | | 1.6E + 15 | 6.7 | 1.3E + 15 | 4.9 |
| Total mass of impactors (based on the 800 Ma spike model) | | | | | 2.3E + 15 | 7.5 | 1.8E + 15 | 5.5 |

[a]craters with an age identical to Copernicus in the constant model.
[b]craters with an age identical to Copernicus in the 800 Ma spike model.

**Fig. 4 The Terrain Camera images and their cumulative size-frequency distributions in various regions of Copernicus crater.** The terrain camera images and their cumulative size-frequency distributions from the floor, ejecta area and melt region near the central peak of Copernicus crater.

into external forcing from asteroids that might have driven ecosystems towards larger and increasingly complex organisms after 800 Ma, although further quantitative discussion will be required.

## Methods

**The method of crater size-frequency measurement.** The model ages of the lunar craters were determined by crater size-frequency measurements on ejecta blankets of the craters using image data obtained by the Terrain Camera (TC) onboard SELENE. The lunar explorer SELENE (KAGUYA) was launched on 14 September 2007 and had a nominal observation of 1 year at an altitude of 100 km[47]. The TC is a panchromatic push-broom imager with two optical heads (TC1 and TC2) to acquire stereo data for the entire surface of the Moon with an average resolution of 10 m/pixel. The details of the instruments have been published elsewhere[48,49].

The technique of crater size-frequency measurement has been described in detail in several papers[50–52]. Therefore, we briefly present the procedure. Based on the simple idea that older surfaces accumulate more craters, the relative age is determined by measuring the cumulative crater frequency at a reference diameter (usually > 1 km), where the measurement of the crater size-frequency distribution (CSFD) is carried out with remote sensing image data. The cratering chronology formulated by relating crater frequencies to the radiometric ages of Apollo and Luna samples enables us to convert the crater frequency into an absolute model age.

We used images that are map-projected in a transverse Mercator projection with a resolution of 10 m/pixel at the central meridian of the projection because this projection introduces little distortion in a narrow area. Crater counting was

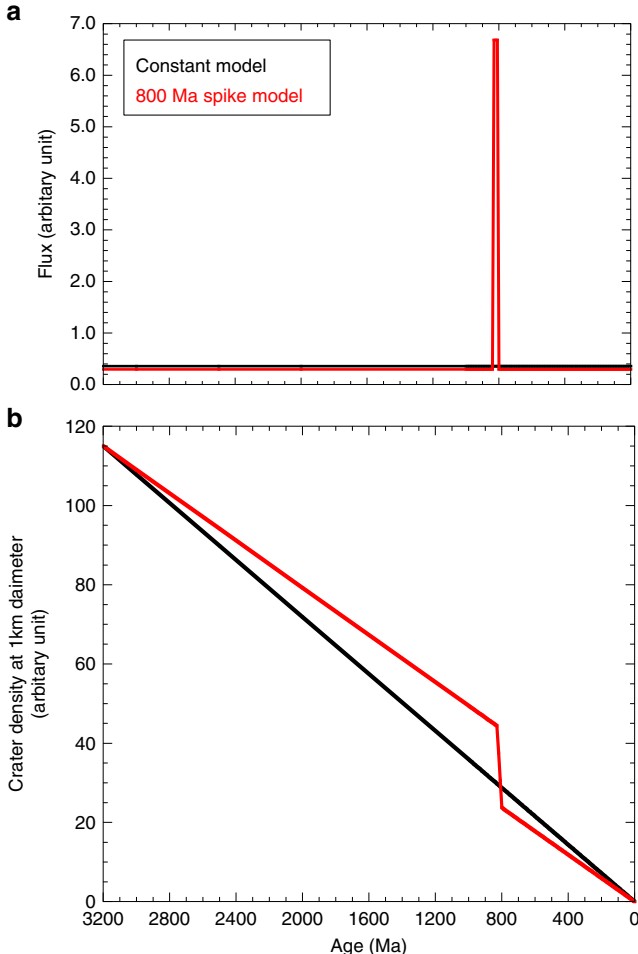

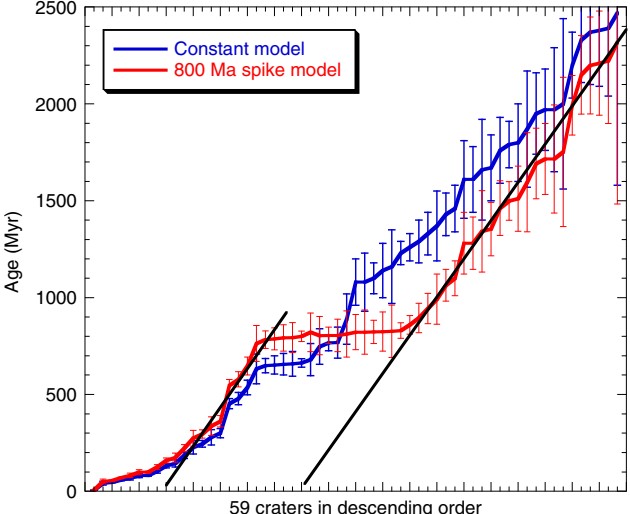

**Fig. 6 The 59 crater ages in descending order.** The estimated ages of 59 craters based on the constant flux model and the 800 Ma spike model are shown in descending order. In both models, it is obvious that the slopes of the lines less than a few million years and ~800 Ma are gentler than those of other eras (e.g., 300–660 Ma and/or >800 Ma), suggesting the recent increase in impact flux forming the large craters (D > 20 km).

and the diameter of craters as follows:

$$D = 1.37\rho_i^{0.22/3}\rho_t^{-1/3}v^{0.44}g^{-0.22}m^{0.26}$$

where $\rho_i$ and $\rho_t$ are the densities of the impactor and the crust and v, m, and g are the impact speed, impactor mass, and gravity, respectively. Assuming a density of near-Earth asteroids (1.29 g cm$^{-3}$ for C-type asteroid Ryugu[15], 1.9 g cm$^{-3}$ for S-type asteroid Itokawa[16] and 2.7 g cm$^{-3}$ for S-type asteroid Eros[17]) and a relative velocity of 20 km s$^{-1}$ for Earth-crossing asteroids to the Moon[18,19], the mass and size of the impactors for 17 lunar craters that coincide with the formation of Copernicus crater are shown in Table 2. It should be noted that these parameters do not affect the conclusion much, even if the density changes from 2.5 to 3.5 g cm$^{-3}$ and the velocity changes from 10 to 20 km s$^{-1}$. Although numbers of smaller craters <20 km in diameter would increase, the observed craters with sizes of 35–93 km are the main contributor to the total mass of bombardment (e.g., masses of the impactors for a D = 10 km crater and a D = 3 km crater are 1/100 and 1/10$^6$ of those for a D = 35 km crater, respectively).

## Data availability
All crater-counting data analyzed during this study are provided in Table 1 and in Supplementary Information.

## Code availability
We used the software tool craterstats[14] to fit the observed crater size distributions to the crater production function and to calculate its errors (http://hrscview.fu-berlin.de/software.html).

**Fig. 5 Two flux models and their cumulative curves of crater numbers with time. a** illustrates a conventional constant model and an 800 Ma spike model, and **b** illustrates their cumulative curves of crater numbers as a function of time, where total counts of craters older than 3.2 billion years are adjusted to be the same in both models.

performed within the continuous ejecta out to one crater radius from the crater rim. In this study, we eliminated obvious secondary craters based on their morphological characteristics, such as chain craters, elliptical craters, and clusters.

In general, self-secondary cratering would cause the estimated ages to be older than the actual ages[53,54]. Although self-secondary craters[54,55] may be included in our counting, the effect of self-secondary craters on the age determination is sufficiently small to ignore. The crater frequency (the density of craters larger than 1 km in diameter) of Giordano Bruno, which is the youngest crater among those investigated in this study, is $4.9 \times 10^{-6}$ km$^{-2}$, which is considered the maximum flux value of self-secondary craters if the age is zero (in actuality, this value is the summation of both self-secondary craters and cumulative craters after Giordano Bruno formation). Thus, the maximum contribution of self-secondary craters is ~100 times smaller than that corresponding to 660 Ma ($5.5 \times 10^{-4}$ km$^{-2}$), suggesting that self-secondary crates should be <1/100 of the small craters counted on the continuous ejecta for craters with a model age of 660 Ma. Therefore, self-secondary craters do not affect the discussion of this paper.

We used the lunar standard CSFD and the cratering chronology model proposed by Neukum and co-workers[55–57] to obtain the absolute model age from the CSFD measurement. More recently, the standard CSFD and the chronology model have been updated[52,57]. However, in the diameter range (a few hundred metres to a few tens of metres) and the age range (0.1 Ga) used in this study, the differences between the models are insignificant[42] and produce only slight differences in the estimated ages[58]. Therefore, we adopted the Neukum model[55] in this study. The model ages from crater counts are principally limited by the statistical error. The statistical error of individual data points in our crater frequency measurements is mostly <30% (1 sigma).

**Estimation of the size and mass of the impactor.** According to Shibaike et al.[14] (originally from Abramov et al.[59]), the mass of the impactor can be estimated from the density of the impactor, the density of the crust, the velocity of the impactor

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

## Acknowledgements

We wish to express their sincere thanks to all the members of the SELENE project. We thank Prof. Tajika from the University of Tokyo for useful discussion. This work was partly supported by Japan Society for the Promotion of Science (JSPS) KAKENHI Grants (No. 18H01269).

## Author contributions

K.T. conducted the entire research theme and wrote the paper. T.M. and M.K. contributed all of the analysed data and wrote the method.

## Competing interests

The authors declare no competing interests.
