## [Peer Review File · Nature Communications]

Reviewers' comments:

Reviewer #1 (Remarks to the Author):

This is another example of a few recent papers that demonstrates the use of lunar crater data to help explain terrestrial processes. The paper is innovative and clearly written. I find no major flaws in the data, models, or arguments, and thus recommend to publish this paper almost "as is".

A few minor points: one should be careful not to confuse mass flux onto the Earth (or Moon) in general with impact frequency. The works by (e.g.) Schmitz and others that are cited refer mainly to meteorites and small objects (even osmic dust), which can rain down over much longer time periods - examples include the late Miocene ET dust shower (the "Veritas" event) or the late Eocene ET accumulation - see e.g: [https://doi.org/10.1130/2009.2452\(03\)](https://doi.org/10.1130/2009.2452(03))

Or even trhe KT comparison, In the other cases there werre long accumulations of ET dust, and impacts only at certain times; in the case of the KT only a big impact but not much ET dust (as seen by He-3 studies). So just mass accumulation is one thing, impacts another. But in the present case, the lunar record indicates impacts as well. It might be worth loking more carefully at possible impacts during the Snowball events on Earth.

The authors cite reference 25 (by the way, updated page numbers: Meteoritics & Planetary Science 54, Nr 10, 2273–2285 (2019)) and note that one impact alone may or ma not influence climate (the conclusions in ref 25 are actually: we can't say because we did not run a climate model, but there is a lot of dust that goes into the atmosphere and so thee might be albedo changes); and also that study was not so much about starting but about possibly ending a Snowball phase. Maybe some fine-tuning in the discussion could be done.

I look forward to seeing this work in print.

Sincerely,

Christian Koeberl

(Natural History Museum Vienna & University of Vienna, Austria).

Reviewer #2 (Remarks to the Author):

Terada et al. provided a very interesting idea based on the estimation of crater age on the Moon. They found that there was a cluster of craters with an age of 658 ± 16 Ma, which coincides with one of a few Neoproterozoic glaciations on the earth, namely the Marinoan glaciation. Interestingly, only the Marinoan glaciation is characterized by increased concentrations of Ir (Bodiselsch et al., 2005 Science, v. 308, p. 239-242), whereas other Neoproterozoic glaciations lack the elevated Ir concentrations as well as other PGEs anomalies (Bodiselsch et al., 2005 Science, v. 308, p. 239-242; Ivanov et al., 2013 Geology, v. 41, p. 787-790). I think this should be highlighted in the text. At present this information is not obvious for the reader.

The crucial for the submitted paper is the correctness of Copernicus crater age estimation. I'm not an expert on the Moon cratering record and leave this part to someone, who is familiar with the topic. If this part is correct, then I suggest that the paper should be accepted for publication after minor revision. Below I provide some comments and suggestions.

Lines 40-43: This sentence should be rewritten because in the present form it looks misleading. For example, the Chicxulub impact likely did not cause the Cretaceous-Paleogene mass-extinction, because the impact preceded the mass-extinction by hundred thousand years, whereas the mass-extinction followed immediately after the most voluminous eruptions in the Deccan flood basalt province. Probably Chicxulub impact triggered voluminous eruptions and volcanism killed (e.g. Richards et al., 2015 GSA Bulletin, v. 127, p. 1507-1520). The latter hypothesis is, at least, should be mentioned, so the reader won't get misled. Impact and flood basalt hypotheses are competing with each other. There is a great number of publications, that the mass-extinctions, including two mentioned in the text among many others, were coeval with the flood basalt eruptions. I suggest the text should sound like "Since the 541 Ma Cambrian biodiversity explosion, mass extinction events have occurred at least five times (so-called Big 5 events), and extra-terrestrial impacts are considered as potential cause of some of them (e.g. Late Triassic and Cretaceous-Paleogene extinctions, refs) competing with the flood basalt eruptions related hypotheses (refs).

Lines 98-99: I do not understand what authors mean by chemical co-evolving of the Earth-Moon system over 4.5 Ga? The earth is evolving up to now via subduction and multiple recycling (i.e. plate tectonics), whereas there is no plate tectonics on the Moon.

Line 107: Reference 18 is apparently out of place here, which discusses PGE signal in one of the Precambrian sedimentary records.

Line 112: Cryogenian was not completely cold period. Actually, your event (if the age estimated correctly) corresponds within age uncertainty only with Marinoan glaciation at ca 635 Ma.

General comment: There is much confusion about capitalizing 'the earth' and using or omitting definite article 'the' before it. I personally like Sengor's argumentation
<https://www.geosociety.org/gsatoday/archive/27/3/pdf/i1052-5173-27-3-19.pdf>

Reviewer #3 (Remarks to the Author):

Here is my review of “Cryogenian asteroid shower on the Earth—Moon system revealed by the lunar orbiter KAGUYA” by Terada et al.

I am going to present this as a “good news, bad news” review.

The good news is that there is a lot to like in this paper, and I think the authors have found evidence for an impact spike on the Moon that may prove to be compelling.

The bad news is that I am unconvinced by their arguments that their putative impact spike occurs at either of the ages suggested in the paper (e.g., near 470 Myr or 660 Ma). Given that most of their paper concentrates on those two likelihoods, I think it needs to be completely rewritten.

Instead, I would argue a much stronger paper would come from editing those possibilities out of the paper in favor of 800 Ma, the time when I suspect their impact spike really did take place (see below).

Note that if the authors wish to continue to discuss the 470 Myr or 660 Ma spikes in addition to an 800 Myr spike, that is their option, but I am not sure the paper would still be appropriate for Nature Communications. Such a manuscript would present the reader with too many options -- though ultimately this would be up to the editor to decide how to proceed.

Here are my comments on specific issues.

1) The age of Copernicus.

A critical issue with lunar chronology is that almost no terrains have sample ages over the last 3.2 Gyr. One of the main regions that does have an age is Copernicus crater, one of the lynchpin craters in the Terada et al. hypothesis. Apollo 12 landed on a ray from Copernicus, and the astronauts observed high albedo material in several locations (Stoffler and Ryder 2001). The age of this returned lunar material is ~800 Myr from both Ar-Ar and U, Th-Pb (see references in Stoffler and Ryder 2001). The authors also do not discuss that there is an 800 Ma spike in impact spherule ages from Apollo 12, 14, 16, and 17 samples as well (see Zellner et al. 2009; reference below). Therefore, if their putative impact spike is real, the most likely circumstance is that it occurred ~800 Ma. At the very least, I believe the authors need to consider this idea in their paper.

The paper instead concentrates on putative Copernicus ages that, in my opinion, have far less support than from the direct Apollo 12 sample evidence (e.g., lines 270-280).

The rationale that Copernicus is ~660 Ma is from crater counting age alone. It assumes that crater spatial densities on Apollo 12 terrains, when divided by the age of 3.2 Ga from Apollo 12 samples, yield an average impact flux that can perfectly date crater spatial densities on Copernicus terrains. However, crater counts on Apollo 12 terrains and Copernicus surfaces both have substantial error bars, and their age assumption does not discuss this issue. There is also no concentration of lunar impact spherule ages at 660 Ma.

I do not find a single 470 Ma zircon age in Apollo 12 material to be a strong enough rationale to argue that Copernicus itself was 470 Ma, though I do favor the idea of some kind of impact spike at that time (see below). At every Apollo site, there is evidence that nearby craters have launched ejected material onto terrains where the astronauts walked and retrieved samples. Some ejecta has also heated materials on those terrains (e.g., see recent papers by C. Crow, where lunar zircons have age disturbances by heating events over the last few hundreds of Myr). Similarly, lunar impact spherule data also shows every Apollo site has a range of spherule ages, with most of this material presumably delivered there as ejecta. This seems to be the most likely origin for the single 470 Ma zircon.

2) There is possible additional lunar evidence for an impact spike at 800 Ma.

Related to the discussion above, I encourage the authors to read the paper:

Zellner, N. E. B., Delano, J. W., Swindle, T. D., Barra, F., Olsen, E., & Whittet, D. C. B. (2009a). Evidence from $^{40}\text{Ar}/^{39}\text{Ar}$ ages of lunar impact glasses for an increase in the impact rate ~800 Ma Ago. *Geochimica et Cosmochimica Acta*, 73(15), 4590–4597.

The first sentence of the abstract is: “Geochemical and $^{40}\text{Ar}/^{39}\text{Ar}$ data on nine impact glasses from the Apollo 14, 16, and 17 landing sites indicate at least seven distinct impact events with ages ~800 Ma.” They conclude as follows: “... and in the context of diverse compositional range and sample location, there is a suggestion that there may have been a transient increase in the global lunar impact flux at ~800 Ma. Therefore, the Copernicus impact event could have been one of many. If correct, there should be evidence for this increased impact flux around 800 Ma ago in the age statistics of terrestrial impact samples.”

I have personally been looking for evidence of an impact spike near 800 Ma since this paper came out. Nicole Zellner stands behind it, and I think it is an intriguing hypothesis.

In my opinion, the Terada et al. results would seem to provide evidence that could explain Zellner et al. (2009), but they would have to assume Copernicus formed 800 Ma, as suggested by Apollo samples. I will have more to say on this issue below.

3) There is no evidence for the Terada et al. impacts spikes in Mazrouei et al. (2019)

The structural framework of Terada et al., and some of its main ideas, are closely associated with this paper:

Mazrouei, S., R. R. Ghent, W. F. Bottke, A. H. Parker, T. M. Gernon. 2019. Earth and Moon impact flux increased at the end of the Paleozoic. *Science* 363, 253-257.

It is therefore curious that Terada et al. do not reference Mazrouei et al. (2019). In my opinion, this needs to be corrected in the next version of their paper.

With that said, many of the conclusions of Terada et al. – as written – go against the results from Mazrouei et al. (2019). Now, it is possible that Mazrouei et al. (2019) is incorrect, but I will argue here that a more interesting possibility is that the two papers are complementary.

For reference, Mazrouei et al. (2019) derived ages for all non-polar $D \geq 10$ km lunar craters that were less than roughly ~ 1 Ga. This was done by using temperature data derived from LRO's Diviner thermal radiometer to determine the abundance of large impact ejecta fragments, or "rocks", surrounding those craters. They found the production rate of $D \geq 10$ km lunar craters was 2-3 times higher over the last ~ 290 Myr than it had been over the previous 700 Myr. Notably, they also found a comparable increase for 38 terrestrial craters with $D \geq 20$ km and ages < 650 Ma, 85% which are on geologically stable cratons.

A key line of support that the Mazrouei et al. (2019) results are reasonable are that they predict the lunar and terrestrial crater records are similar to one another. If their lunar results were incorrect, perhaps because they used a faulty method, there should be little correspondence.

Critically, Mazrouei et al. (2019) shows no evidence for an impact spike near 660 Ma in the lunar record, one of the suggested times of the Terada et al. impact spike. The terrestrial record is more difficult to evaluate at 660 Ma, mainly because there is evidence for massive erosion near that time, but work on asteroid families indicates that impact "spikes" often produce a substantially tail of impacts after the main spike is complete (i.e., objects created by an asteroid breakup near a dynamical resonance will evolve into the resonance by the Yarkovsky effect). There is no evidence in the terrestrial record for such an impact tail. Therefore, I am skeptical that there was an impact spike at this time.

Terada et al. try to move their putative spike to 470 Myr, but that seems unlikely to me for a number of reasons.

Note that both the terrestrial and lunar crater records show intriguing but somewhat limited crater evidence for an impact spike at 470 Ma, the likely time of the L-chondrite breakup event. This can be potentially seen in the Mazrouei et al. (2019) results, though the signature is mainly in $D > 10$ km craters than $D > 20$ km craters. If the Gefion family is indeed the source of the L chondrite shower, as discussed by Nesvorný et al. (2009) (i.e., see the match between their model results and the cosmic ray exposure ages of chromite grains from 470 Ma samples), we would probably expect the Earth and Moon to be dominated by smaller impactors (see next paragraph). This does not fit the Terada et al. signature which includes Copernicus.

The reason for small impactors is that the 5:2 resonance closest to the Gefion family delivers relatively few impactors to the Moon (i.e., about 1 in 100,000 inserted into the resonance; see Supplemental Materials from Bottke et al. 2006; Nature). We understand the Gefion family evolution well enough from Nesvorný et al. (2009) to know that it did not inject 100,000 Copernicus-class impactors into the resonance.

Consider as well that for every Copernicus event that occurs on the Moon, there should be 20 similar-sized events on Earth. That is enough that it is statistically likely some should have occurred on stable terrestrial cratons, which make up $\sim 10\%$ of the Earth's surface. There is no

evidence for such major events at that time on Earth on those cratons. Even if the big putative impactors managed to avoid stable cratons, we would expect some impacts to have produced impact spherule beds that would have been found by terrestrial geologists (e.g., see Glass and Simonson 2012). Finally, there is some limited evidence that the impactor that made Copernicus was a carbonaceous chondrite, not an ordinary chondrite. This would work against the idea that Copernicus came from the L-chondrite breakup event (see various papers by Morgan et al.).

Now, if the impact spike suggested by Terada et al. were moved to 800 Ma, as suggested above, I think it would be much more consistent with the results of Mazrouei et al. (2019). That age was very close to their limit of resolution (i.e., rock abundance signatures go to the background after ~ 1 Gyr). Indeed, many of the craters with crater spatial densities similar to Copernicus in the Terada et al. data are found to have limited rock abundance signatures. So, finding that some craters smaller than Copernicus were near the age of Copernicus would not be a big surprise.

4) The asteroid families suggested as sources for the 650 Ma spike are incapable of doing so.

The paper suggests that the Agnia or Hansa families could have produced the asteroid shower, and references Spoto et al (2015). This gets into issues that are beyond this paper, but the ages of these families do not line up with other dynamical age-dating work for those families. For example, Vokrouhlicky et al. (2006) showed that Agnia was probably ~ 100 Ma. The age of Hansa is unpublished, but we can predict from its shape that is probably closer to 300 Ma (e.g., see method in Walsh et al. 2013 for an example).

Even if the Spoto et al. ages were correct, though, the listed families are not sizable enough or well positioned enough to produce the kind of spike suggested by the authors. Consider that to make an impact spike, one needs to explain Copernicus, which is a 93 km lunar crater. Assuming a crater to projectile diameter ratio of 20, Copernicus was made by a ~ 5 km projectile.

The Agnia family is located near the 5:2 resonance, where the probability of a projectile hitting the Moon is $\sim 10^{-5}$ (e.g., see Supplemental Materials from Bottke et al. 2006; Nature). That means that to get one Copernicus impact, Agnia would need to inject 100,000 $D > 5$ km bodies into the 5:2 resonance. As shown in Vokrouhlicky et al. (2006), this family is incapable of doing this; very few $D > 5$ km bodies are even near the 5:2 resonance. The Hansa family is located at high inclinations near the 3:1 and 8:3 resonances. This region also has a difficult time producing impactors, and Hansa is too small to have injected so many $D > 5$ km bodies into resonances over a short time.

In summary, the Terada et al. putative impact spike at 650 Ma has no obvious source in the asteroid belt, and it needs one.

However, there is a family that can potentially produce an impact spike at ~ 800 Ma. See:

Bottke, W. F. and 9 co-authors. 2015. In Search of the Source of Asteroid (101955) Bennu: Applications of the Stochastic YORP Model. *Icarus* 247, 191-271.

They argued that the Eulalia family had an age of 830 [+370, -100] Ma. Moreover, when it disrupted, a large share of the sizable family was directly injected into the 3:1 resonance at low inclinations. This disruption almost certainly produced an impact spike, which has to be placed somewhere in time. It could make sense that it is linked to the formation of Copernicus; that was certainly the opinion of the lead author at the time of its writing. It is also a carbonaceous chondrite family, so it could presumably produce the limited compositional constraints we have on Copernicus crater (e.g., Morgan et al. papers)

5) Issues with methods for the crater counts and their ages

Overall, the basic crater counts look reasonable, but I have several issues with the work that need to be discussed or corrected. I have grouped all of these issues, but each one should be treated as meaningful as points #1-#4 above.

5a) Computing ages:

Terada et al. acknowledge that the different terrains can have different properties and so avoid melt pools (this is good). But then they do not account for the fact that the Neukum chronology wasn't built for ejecta but for harder terrains. I believe Carolyn van der Bogert discusses this in her recent paper (Icarus, 2017, p. 49-63), and possibly even how to correct for it. The problem is the authors do not seem to know about these issues (or at the least do not acknowledge these issues).

This, however, is a smaller issue compared to the fact that the calculated errors on their crater ages have poor accuracy. I do not think the authors are using the latest version of "craterstats", or if they are, they are not using as others in the crater field use it. The errors on their crater counts seem quite small, as was often the case for the older version (and even to an extent the newer version) of craterstats. Also, this introduces a bias into their probability age distribution (Fig. 3A) because the errors will always be smaller on the younger craters than the older ones, making "spikes" appear because the gaussians are thinner. Crater experts that I know often assume a minimum of 0.1 Ga for errors even on the youngest crater ages. Even this is probably too small, but does remove the bias.

Related to this, their discussion of the uncertainties due to only statistics in the Methods section is wrong.

5b) Statistical significance of their data

The authors perform no statistical analysis to make sure their "spikes" are real. They just say 7 out of 59 is significant. This needs to be shown statistically.

For example, one could create a Monte Carlo code where there are 59 craters formed at different ages. One could assign age errors to every crater formed. One could then determine how often

an age spike was created by chance. The larger the error bars, the more often spikes would take place by chance. This would give the reader some feeling for whether what they are seeing is a fluke or something meaningful.

In addition, the authors seem to be using bins of 50 Myr for their histograms, but this is too small given the (realistic) errors on their crater ages.

5c) Including a spike in their analysis

The authors do not discuss the apparent contradiction in their method.

They assume the small crater production rate is constant so they have the means to get the ages of the big craters by superposed crater counts. If the impact flux was constant across all sizes, though, it would be impossible to see a spike.

The only way their method can find a spike is to assume the production of big craters and small craters are decoupled from one another.

This issue is not discussed in this paper, but it needs to be. The authors could look at the Supplemental Material in Mazrouei et al. (2019) for an example of how this was addressed.

5d) Ignoring the well-cited age of Copernicus

To get their preferred ages for their craters, the authors ignore the well-cited age of Copernicus of 800 Ma and simply modify their lunar crater chronology to get what they prefer. This assumption is poorly justified, other than the authors want their putative spike to line up with the events they choose.

I believe they need to try Copernicus at the accepted age first and see what the results are, then move it around within the range argued for by others and see if and how that changes the results.

6) Impactor mass calculation

The authors have no reference for the bulk density of their impactors nor their impact velocities.

It is unclear to me why an asteroid would have a bulk density of 3 g cm^{-3} when no asteroid yet studied has such a bulk density (e.g., most S-types are between $1.9\text{-}2.7 \text{ g cm}^{-3}$; C-types are closer to 1.3 g cm^{-3})

Impact velocities for the Moon have been discussed in many references. Most projectiles are not hitting the Moon at 15 km/s , so it is not clear why this velocity was chosen.

7) There is little evidence that impacts caused the Snowball Earth event at 650 Ma.

The authors discuss in the text how there is little evidence that impacts caused the Snowball Earth event at 650 Ma. Given the discussion above, their own text provides a further rationale that the paper should move away from linking the two.

8) Other issues

I do not believe the authors can make a strong case for an impact spike at ~660 Ma. I also argue their rationale to move their impact spike to 470 Ma is poorly supported. Perhaps another way of saying this is that while there is evidence for some kind of impact spike at 470 Ma, I am skeptical its signature is associated with Copernicus and the other craters discussed by Terada et al.

For this reason, much or all of their text discussing evidence for a 470 Ma impact spike is not germane to the topic at hand. I argue the authors need to find better ways to argue their impact craters are really 470 Ma or all of this text could be eliminated for a revised paper.

Point by point response

Reply to Reviewer #1:

The authors cite reference 25 (by the way, updated page numbers: *Meteoritics & Planetary Science* 54, Nr 10, 2273–2285 (2019)) and note that one impact alone may or may not influence climate (the conclusions in ref 25 are actually: we can't say because we did not run a climate model, but there is a lot of dust that goes into the atmosphere and so there might be albedo changes); and also that study was not so much about starting but about possibly ending a Snowball phase. Maybe some fine-tuning in the discussion could be done.

According to the suggestion, we modified the page number of reference 25.

We also carried out fine-tuning in the discussion (around line 234-236).

Reply to Reviewer #2:

Terada et al. provided a very interesting idea based on the estimation of crater age on the Moon. They found that there was a cluster of craters with an age of 658 ± 16 Ma, which coincides with one of a few Neoproterozoic glaciations on the earth, namely the Marinoan glaciation. Interestingly, only the Marinoan glaciation is characterized by increased concentrations of Ir (Bodiselič et al., 2005 Science, v. 308, p. 239-242), whereas other Neoproterozoic glaciations lack the elevated Ir concentrations as well as other PGEs anomalies (Bodiselič et al., 2005 Science, v. 308, p. 239-242; Ivanov et al., 2013 Geology, v. 41, p. 787-790). I think this should be highlighted in the text. At present this information is not obvious for the reader.

After the correction, we concluded that Asteroid shower occurred the 800 Ma, just before the large-scale glaciation (**Kaigas-Sturtian glaciation (730-700 Ma)** and Marinoan glaciation (650 to 635 Ma)).

According to the suggestion, we highlighted that only the Marinoan glaciation is characterized by increased concentrations of Ir, whereas other Neoproterozoic glaciations lack the elevated Ir concentrations as well as other PGEs anomalies. (around line 220-223)

The crucial for the submitted paper is the correctness of Copernicus crater age estimation. I'm not an expert on the Moon cratering record and leave this part to someone, who is familiar with the topic. If this part is correct, then I suggest that the paper should be accepted for publication after minor revision. Below I provide some comments and suggestions.

According to the suggestion, we carefully investigated the Copernicus crater age estimation based on the “conventional” constant flux model, again, and confirmed that our counting method is correct and consistent with other previous works. We mentioned them around line 102-118.

Finally, we regard the age of the Copernicus as 800 Ma based on the radiometric age of Apollo glass spherules.

Lines 40-43: This sentence should be rewritten because in the present form it looks misleading. For example, the Chicxulub impact likely did not cause the Cretaceous-Paleogene mass-extinction, because the impact preceded the mass-extinction by hundred thousand years, whereas the mass-extinction followed immediately after the most voluminous eruptions in the Deccan flood basalt province.

Probably Chicxulub impact triggered voluminous eruptions and volcanism killed (e.g. Richards et al., 2015 GSA Bulletin, v. 127, p. 1507-1520). The latter hypothesis is, at least, should be mentioned, so the reader won't get misled. Impact and flood basalt hypotheses are competing with each other. There is a great number of publications, that the mass-extinctions, including two mentioned in the text among many others, were coeval with the flood basalt eruptions.

I suggest the text should sound like "Since the 541 Ma Cambrian biodiversity explosion, mass extinction events have occurred at least five times (so-called Big 5 events), and extra-terrestrial impacts are considered as potential cause of some of them (e.g. Late Triassic and Cretaceous-Paleogene extinctions, refs) competing with the flood basalt eruptions related hypotheses (refs).

According to the suggestion, we have replaced the previous sentences with "and extra-terrestrial impacts are considered as potential causes of some of them (e.g. Late Triassic and Cretaceous-Paleogene extinctions) competing with the flood basalt eruptions related hypotheses". (line 44-45)

Lines 98-99: I do not understand what authors mean by chemical co-evolving of the Earth-Moon system over 4.5 Ga? The earth is evolving up to now via subduction and multiple recycling (i.e. plate tectonics), whereas there is no plate tectonics on the Moon.

I meant that lunar meteorites come from Moon to Earth and terrestrial oxygen was stripped and transported to the Moon and implanted the lunar soils, which was my discovery (Terada et al. Nature Astronomy 2017). But I deleted this sentence because it might be redundant.

Line 107: Reference 18 is apparently out of place here, which discusses PGE signal in one of the Precambrian sedimentary records.

Reference 18 is wrong, here. I deleted.

Line 112: Cryogenian was not completely cold period. Actually, your event (if the age estimated correctly) corresponds within age uncertainty only with Marinoan glaciation at ca 635 Ma.

We replace "Neoproterozoic Cryogenian period" with "Marinoan glaciation".

Reply to Reviewer #3:

Here is my review of “Cryogenian asteroid shower on the Earth—Moon system revealed by the lunar orbiter KAGUYA” by Terada et al. I am going to present this as a “good news, bad news” review. The good news is that there is a lot to like in this paper, and I think the authors have found evidence for an impact spike on the Moon that may prove to be compelling. The bad news is that I am unconvinced by their arguments that their putative impact spike occurs at either of the ages suggested in the paper (e.g., near 470 Myr or 660 Ma). Given that most of their paper concentrates on those two likelihoods, I think it needs to be completely rewritten. Instead, I would argue a much stronger paper would come from editing those possibilities out of the paper in favor of 800 Ma, the time when I suspect their impact spike really did take place (see below). Note that if the authors wish to continue to discuss the 470 Myr or 660 Ma spikes in addition to an 800 Myr spike, that is their option, but I am not sure the paper would still be appropriate for Nature Communications. Such a manuscript would present the reader with too many options --though ultimately this would be up to the editor to decide how to proceed.

At first, we appreciate for your great instructive and pertinent comments. Recently we found that KAGUYA observed the constant carbon emitting from the total lunar surface (larger than in-flux by solar wind origin and current micrometeorite' flux), suggesting that lunar surface has been contaminated by volatile element (Yokota et al. Science Advances, in press). Taking reviewer's pertinent comments and our new observation into consideration, we have changed the interpretation the timing of sporadic asteroid showers, possible 800Ma.

Here are my comments on specific issues.

1) The age of Copernicus.

I didn't know the Zellner, et al. (2009a). Now, we agree that the age of sporadic crater formation, including Copernicus, is 800 Ma.

2) There is possible additional lunar evidence for an impact spike at 800 Ma.

Again, I didn't know Zellner et al. (2009a). It is so pertinent and compelling to consider that clustered age of glass spherules from various landing sites is related to the clustered age of craters formation in this study because the probability is too low (0.68% by accident). Now, we agree that the age of sporadic crater formation, including Copernicus, is 800 Ma.

3) There is no evidence for the Terada et al. impacts spikes in Mazrouei et al. (2019)

According to the reviewer's suggestion, we carefully read Mazrouei et al. (2019) and relevant papers of numerical simulation of the Gefion family. Now, we agree that sporadic crater formation 800 Ma is reasonable in many senses (volatile-rich Copernicus, Mazrouei, flux induced from Gefion family) which reviewer#3 raises.

Also during our consideration, we found that our data also suggest that production rate of lunar craters (>20km) younger than 300 was higher than those of 300-650 Ma, which is consistent with Mazrouei et al. (2019)^[24]. So we added a new figure and explanation (around line 189-193), which illustrates the age frequency distribution as follow.

4) The asteroid families suggested as sources for the 650 Ma spike are incapable of doing so. < Partially omitted> However, there is a family that can potentially produce an impact spike at ~800 Ma.

According to the instructive comments, we realized that 800Ma is more reasonable for the reported phenomena than 660 Ma. Now, we have modified the interpretation of the sporadic formation of lunar craters.

5) Issues with methods for the crater counts and their ages

Overall, the basic crater counts look reasonable, but I have several issues with the work that need to be discussed or corrected. I have grouped all of these issues, but each one should be treated as meaningful as points #1-#4 above.

5a) Computing ages:

The Neukum chronology was built based on the crater size-frequency measurements on basin ejecta and mare basalts for the age range older than 3.0 Ga but was also used crater count data on ejecta deposits of young craters such as Tycho, South Ray, and North Ray for the age range younger than 1.0 Ga. As the reviewer pointed out, the target strength affects on crater size and the selection of counting area is important for age estimates. We performed crater size-frequency measurements on the ejecta deposits of the craters, not melt ponds. This is consistent with using the Neukum chronology. Although we refereed Bogert (Icarus, 2017, p. 49-63), we highlighted it as reference 13 (line 75). We believe that the version of “craterstats” is not crucial for this discussion.

As for the estimation of age, an error is based on the counting statistics (that is, root N), which is a usual approach for crater chronology. As the reviewer pointed out, the small errors of young craters show misleadingly “apparent sharp” peak in the probability distribution diagram. Therefore, we added a histogram with an age bin size of 0.1 Gyr to Fig. 3.

5b) Statistical significance of their data

Monte Carlo simulation shows that the possibility that seven of 59 crater formation happen at the same time (for 50Ma from 630 to 680Ma) by chance is 0.69%, where the 54S161E crater (747 ± 92 Ma) is masked because it is obviously outlier with large uncertainties. If include 4S161E crater (747 ± 92 Ma), the possibility that eight of 59 crater form during the 100 Ma by chance is 7%. According to the suggestion, we added this statement.

In addition, the authors seem to be using bins of 50 Myr for their histograms, but this is too small given the (realistic) errors on their crater ages.

According to the suggestion, we modified the bin-width.

5c) Including a spike in their analysis.

Yes, the comment is so pertinent. A “conventional” constant flux model assumes that a large crater production rate (>20km) and a small crater production rate (0.1-1km) are decoupled, whereas, for the case of the 800 Ma spike model, the obtained crater production rate (>20km) and a small crater production rate might be coupled. However, the estimated total masses are NOT so significantly changed ($(1.3-1.6) \times 10^{15}$ kg for constant flux model and $(1.8-2.3) \times 10^{15}$ kg for the 800 Ma spike model), because the Copernicus crater is dominant among the 8 craters coincidence (by constant model) and 17 craters coincidence (by spike model). Therefore, the latter discussion on total mass estimation of the impactor is not affected by the choice of the flux models with/without spikes. These discussions are very important, so we added them around line 181-187.

5d) Ignoring the well-cited age of Copernicus

Now, we have modified the interpretation of the sporadic formation of lunar craters and agreed that the age of Copernicus is 800Ma.

6) Impactor mass calculation

We added the references for the bulk density of their impactors nor their impact velocities. And we replaced the velocity of 20 km/sec based on the Feuvre et al. (2008) and or Ito and Malhotra (2010) and recalculated.

7) There is little evidence that impacts caused the Snowball Earth event at 650 Ma.

Our new conclusion is the asteroid shower 800 Myr ago. So, we have modified that the meteoroid shower might have occurred just before the Snowball Earth period.

8) Other issues

According to the above mentioned considerations, we have deleted the discussion on 470 Ma asteroid shower based on the observation of the lunar crater.

Thus, we have responded the all issues that the editor and reviewers raised.

Now, we believe that it will now be acceptable for publication in Nature Communication.

Sincerely yours,
Kentaro Terada

REVIEWERS' COMMENTS:

Reviewer #2 (Remarks to the Author):

I read with interest the revised m/s. From my side, I'm satisfied how authors modified their text.

Alexei V. Ivanov

Reviewer #3 (Remarks to the Author):

Here is my second review of “Cryogenian asteroid shower on the Earth—Moon system revealed by the lunar orbiter KAGUYA” by Terada et al.

I appreciate the work the authors have done to revise their paper, and I am pleased that they adopted my suggested changes. I find this version to be much more scientifically satisfying, given that their hypothesis now lines up with other constraints in the literature.

I can recommend it for publication, with some modest revisions.

The major comment I have on this version of the paper is not scientific, but rather concerns the readability of the paper. It could use the services of an editor that could make the document flow better. It is very choppy from an English perspective, and contains a number of passages I would personally modify (e.g., lines 133-166 is one extremely long paragraph; it uses the word “Besides” several times; etc.).

The Nature copy editors may do this themselves, but someone does need to edit the paper prior to publication.

Here are my specific comments, most which are easy fixes:

Comments:

Lines 46-61. I do not understand the rationale to include a paragraph on the impact spike from 470 Ma here if you are going to argue that an impact spike takes place at 800 Ma. It seems like leftover component from the first round of the paper.

Lines 84-88. I am fine with using a Monte Carlo code here, but there is no explanation of the assumptions or methods used. More needs to be said here or in the methods about what was done.

Lines 98-99. “estimated to be $(1.3-1.6) \times 10^{15}$, corresponding to a 10-13 km in diameter”
Missing a word.

Lines 135-136. “Based on the temperature of large impact ejecta of which carter size is larger than 10 km in diameter”. Crater is spelled incorrectly. This also occurs on line 186.

Line 156. “This disruption certainly could have produced an impact spike on somewhere.”. This is too vague (i.e., somewhere).

Lines 161-162. “the scenario that may have been formed by a cometary nucleus, 4 km in diameter based on geochemistry of the 12033 breccia [29]”.

I am confused. I looked at the Milliken and Li paper referenced here, but I see nothing in the paper regarding a comet impactor, nor anything discussing the 12033 breccia. The paper mainly discusses water signatures in magmas and pyroclastic deposits on the Moon -- and the potential that they are derived from the lunar interior.

Moreover, the paper says this, “None of the observed regions of excess water are clearly associated with impact craters or ejecta, ruling out an origin by impact delivery.”

Is this the reference you wanted to use here, or did you want to use Li and Milliken?

Lines 163-166. I agree that the Moon has most certainly been hit by carbonaceous chondrite-like projectiles in the past; that is evitable given the current mix of S-types and C-types in the near-Earth asteroid population. However, the reference cited is very vague, and provides little support for the hypothesis that an impact spike at 800 Ma is associated with the impact of carbonaceous chondrites.

Lines 193-196. I would not use the word “inclination” here. I suspect the authors are referring to the slope of the lines in Figure 6. If so, they should reword things. Inclination is a commonly-used word in celestial mechanics, and so it can lead to confusion when this word is used in a different context.

Lines 240-243. “Recently, Schmitz et al. (2019) [11] noted that the Ordovician meteorite shower should have triggered the mid-Ordovician ice age based on the sizes of the remaining terrestrial craters, of which total amount of meteoroids is estimated to have 1/300 of that of this study

I am confused by this statement. I think it needs to be edited to be clearer. Schmitz et al. discussed the meteoroid flux on the Earth at that time, but I do not recall them saying anything quantitative about how the sizes of craters told them about the flux of bodies at that time. It is a challenging issue to do correctly.

Best regards,

Bill Bottke

We forgot the “track change” in Word. Instead, the revised descriptions are highlighted in yellow.

Point by point response

Reply to Reviewer (Prof. Bill Bottke)

Lines 46-61. I do not understand the rationale to include a paragraph on the impact spike from 470 Ma here if you are going to argue that an impact spike takes place at 800 Ma. It seems like leftover component from the first round of the paper.

I intentionally leaved this paragraph as an example/introduction of asteroid shower, which is broadly accepted idea.

Lines 84-88. I am fine with using a Monte Carlo code here, but there is no explanation of the assumptions or methods used. More needs to be said here or in the methods about what was done.

According to the suggestion, we added a more detail explanation.

Lines 98-99. “estimated to be $(1.3-1.6) \times 10^{15}$ kg, corresponding to a 10-13 km in diameter” Missing a word.

We added “impactor”.

Lines 135-136. “Based on the temperature of large impact ejecta of which carter size is larger than 10 km in diameter”. Crater is spelled incorrectly. This also occurs on line 186.

So sorry for these typo. I replaced carter with crater.

Line 156. “This disruption certainly could have produced an impact spike on somewhere.”. This is too vague (i.e., somewhere).

I replaces “somewhere” with “terrestrial planets and/or the satellites inside the asteroid belt”.

Lines 161-162. “the scenario that may have been formed by a cometary nucleus, 4 km in diameter based on geochemistry of the 12033 breccia [29]”. I am confused. I looked at the Milliken and Li paper referenced here, but I see nothing in the paper regarding a comet impactor, nor anything discussing the 12033 breccia. The paper mainly discusses water signatures in magmas and pyroclastic deposits on the Moon -- and the potential

that they are derived from the lunar interior. Moreover, the paper says this, “None of the observed regions of excess water are clearly associated with impact craters or ejecta, ruling out an origin by impact delivery.” Is this the reference you wanted to use here, or did you want to use Li and Milliken?

I am sorry that it was wrong reference number. I found, No. 28 (Morgan 1973) and No.29 (Milliken and Li, 2017) had been swapped. We modified.

Lines 163-166. I agree that the Moon has most certainly been hit by carbonaceous chondrite-like projectiles in the past; that is evitable given the current mix of S-types and C-types in the near-Earth asteroid population. However, the reference cited is very vague, and provides little support for the hypothesis that an impact spike at 800 Ma is associated with the impact of carbonaceous chondrites.

As suggested by reviewer, these observation (ref 28-30) is NOT strong evidence but supporting. So we use the term “the scenario is harmonized”. I believe, this expression is acceptable.

Lines 193-196. I would not use the word “inclination” here. I suspect the authors are referring to the slope of the lines in Figure 6. If so, they should reword things. Inclination is a commonly used word in celestial mechanics, and so it can lead to confusion when this word is used in a different context.

According to the suggestion, we used the phrase of “the slope of the lines”.

Lines 240-243. “Recently, Schmitz et al. (2019) [11] noted that the Ordovician meteorite shower should have triggered the mid-Ordovician ice age based on the sizes of the remaining terrestrial craters, of which total amount of meteoroids is estimated to have 1/300 of that of this study”. I am confused by this statement. I think it needs to be edited to be clearer. Schmitz et al. discussed the meteoroid flux on the Earth at that time, but I do not recall them saying anything quantitative about how the sizes of craters told them about the flux of bodies at that time. It is a challenging issue to do correctly.

We deleted the description of “of which total amount of meteoroids is estimated to have 1/300 of that of this study”, because it was the remnant of previous version. In previous version, we estimated the terrestrial flux based on size of existing 480 Ma terrestrial craters on the Earth.

Reply to Editor (against the comments in the manuscript)

- We reduced the words of the title.

- We reduced the words of the abstract.
- We inserted “Introduction”.
- I added the results of the current study to the final paragraph of introduction.
- We deleted all of quotation marks (“ and ”).
- We merged “the discussion” with “Results” to “Results and Discussion”- then we added sub-headings.
- We moved the statement of “We used the software tool craterstats¹⁴ to fit the observed crater size distributions to the crater production function and to calculate its errors (<http://hrscview.fu-berlin.de/software.html>)” from **Acknowledge** to “**Code Availability**”.
- We remove “We also appreciate for instructive comments for Prof. Koeberl and two anonymous reviewers” from **Acknowledge**, according to the suggestion.
- We replaced “Competing financial interests” with “**Competing interests**”.
- We added the brief title for all figures, according to the suggestion.

In addition, for the Figure 4, we slightly modified. In previous version, there were two diagrams for crater sizes and densities and one of them was just “cut & paste” from Hiesinger et al. (2012). Since this is redundant and actually not used for discussion in the manuscript, we deleted the Hiesinger’s diagram in Figure-4.

According to the Data availability policy, we added the Supplementary Information, which include all crater-counting data analyzed during this study

Thus, we have responded the all issues that the editor and reviewers raised. Now, we believe that it will now be acceptable for publication in Nature Communication.

Sincerely yours,
Kentaro Terada